# Urbanization Effects on Surface Wind in the Guangdong–Hong Kong–Macao Greater Bay Area Using a Fan-Sector Method

**DOI:** 10.3390/ijerph19063194

**Published:** 2022-03-08

**Authors:** Dong Xia, Huiwen Nie, Lei Sun, Jing Wang, Kim-Chiu Chow, Kwing-Lam Chan, Donghai Wang

**Affiliations:** 1State Key Laboratory of Lunar and Planetary Science, Macau University of Science and Technology, Macau 999078, China; kcchow@must.edu.mo; 2Zhuhai Public Meteorological Service Center, Zhuhai 519000, China; ggqx_nhw@zhuhai.gov.cn (H.N.); ggqx_sl@zhuhai.gov.cn (L.S.); ggqx_wj@zhuhai.gov.cn (J.W.); 3School of Information Technology, Macau University of Science and Technology, Macau 999078, China; klchan@must.edu.mo; 4Guangdong Province Key Laboratory for Climate Change and Natural Disaster Studies, Key Laboratory of Tropical Atmosphere-Ocean System, Ministry of Education, School of Atmospheric Sciences, Sun Yat-sen University, Zhuhai 519000, China; wangdh7@mail.sysu.edu.cn; 5Southern Marine Science and Engineering Guangdong Laboratory (Zhuhai), Zhuhai 519000, China; 6National Observation and Research Station of Coastal Ecological Environments in Macao, Macau Environmental Research Institute, Macau University of Science and Technology, Macau 999078, China

**Keywords:** surface wind speed, urbanization, GBA megalopolis, OMR method, fan-sector method

## Abstract

Surface wind directly affects human life, wind energy utilization, the atmospheric environment, and many other aspects. The Guangdong–Hong Kong–Macau Greater Bay Area (GBA) megalopolis is experiencing an accelerated progress of urbanization, which may result in the change in surface roughness and atmospheric characteristics. In this study, urbanization effects on surface wind speed (SWS) in the GBA megalopolis, particularly Zhuhai, is investigated by using long-term automatic meteorological measurements, ERA5 reanalysis, and nighttime light data. Results of the analysis show that the averaged SWS has decreased significantly at a rate of −0.53 m s^−1^ per decade over the past decades. With the help of observation-minus-reanalysis (OMR) method, which excludes the atmospheric circulation effects, we found that the decrease in SWS is mainly contributed by the increase in surface roughness, which may account for as much as 75.5% of the decrease. In other words, it is the rapid development of urbanization, rather than the change in large-scale circulation, that could be mainly responsible for the decrease over the GBA in the context of the increasing global SWS since 2010. In addition, a fan-sector method is established to quantitatively analyze the correspondences between urbanization and roughness changes. It is shown that the decrease in wind speed due to surface roughness change is significantly related to the increase in the nighttime light index (NLI) averaged over the 3 km upstream fan-sectors. Moreover, their correlation reaches to 0.36 (negative) when only accounting for the samples of NLI greater than 10. In general, the fan-sector method offers an additional option for assessing the urbanization effects on SWS.

## 1. Introduction

As a representative feature of atmospheric circulation, surface wind has a crucial impact on human health, the atmospheric environment, wind energy utilization, and many other aspects. For example, boundary wind conditions are closely related to pollutant dispersion [1] and outdoor human comfort [2] in urban areas. In addition, wind erosion is mainly responsible for widespread desertification and land degradation in arid and semi-arid regions of northern China [3].

Based on meteorological measurements, a noticeable decrease in surface wind speeds (SWS) mainly before 2010 has been revealed by numerous studies across the globe such as China [4,5,6,7,8], North America [9,10], and Europe [11,12], as well as other regions [13,14,15]. The decrease in SWS was possibly attributed to the increased surface roughness due to the changes in land use and land cover. Recently, Wu et al. [16] analyzed the change in SWS around the world during 1973–2005 and found that the most significant decreases occurred in Central Asia and North America at a rate of −0.11 m s^−1^ per decade, followed by Europe, East Asia, and South Asia with a mean linear trend of −0.08 m s^−1^ per decade. For China, a decline in the SWS has also been reported in many studies. Fu et al. [6] found that the SWS over China showed a distinct declining trend from 1961 to 2007. Jiang et al. [5] reported that the annual mean SWS declined at a rate of −0.12 m s^−1^ per decade from 1956 to 2004. Mcvicar et al. [4] identified a pronounced decrease in SWS over central China during 1960–2006, with a higher rate at higher elevations. Based on in situ measurements, Wu et al. [17] revealed a distinct decrease in SWS at a rate of −0.23 m s^−1^ per decade over the Eastern China Plain (ECP) region during 1980–2011.

Conversely, emission of greenhouse gases and changes in land use from anthropogenic activities have a substantial effect on the atmosphere [18], and the former could be responsible for the warming of approximately 1.1 °C above the pre-industrial global temperature (IPCC 6th Assessment Report) [19]. Some studies have also demonstrated that the decrease in the SWS could be attributed to the overall weakening of large-scale circulation [6,20,21,22]. The circulation variations could induce the changes in pressure-gradient force, which is the primary driving force for wind motion [23]. Meanwhile, several studies have also emphasized that the decrease in wind speed was closely related to the change in the surface roughness, which is caused by urbanization, decreases in areas of forestation or pastures, increased vegetation heights due to increased crop production, etc. [24]. For instance, Zha et al. [25] revealed that the increase in surface roughness could be mainly responsible for a decrement of 0.12 m s^−1^ for the SWS over China, as well as decreases of 0.57 and 0.30 m s^−1^ per decade for the wind speed among large and small cities, respectively. Li et al. [24] quantified the surface roughness effects over China during 1979–2015 using the observation-minus-reanalysis (OMR) method and identified an obvious decrease in observed wind speed at a rate of 0.11 m s^−1^ per decade.

Additionally, recent studies have also illustrated that decadal-scale variations of near-surface winds are probably determined by climatic change (i.e., internal decadal ocean–atmosphere oscillations), rather than by the previously hypothesized urbanization and/or vegetation growth. In particular, although global terrestrial stilling (i.e., SWS decline) has previously been confirmed as an established phenomenon during the period of 1978–2010, a recent reversal trend has been identified in 2010 and onward with an increasing rate of 0.24 m s^−1^ per decade (*p* < 0.001), which is three-fold the decreasing rate before the turning point in 2010 [26]. More relevantly, Yang et al. [27] initially reported that Beijing City presented large SWS differences between rural and urban regions during 2008–2010, which however decreased during 2013–2017.

Overall, the SWS change is under the joint efforts from climatic change (large-scale circulation and decadal ocean–atmosphere oscillations) and surface roughness change. The two factors play different roles in determining the SWS variations at different scales (from local to global) and regions (from rural to urban). However, previous studies are generally focused on the large (regional or continental) scale instead of small (local) scale, although the latter has a comparable importance for climatic change assessments.

The Guangdong–Hong Kong–Macau Greater Bay Area (GBA) megalopolis has been experiencing an accelerated progress of urbanization due to rapid social and economic development, which meanwhile may result in the changes of land use. However, despite a high level of urbanization, there was almost no relevant research for the GBA megalopolis. To investigate the effects of urbanization on the SWS, the analysis of the present study will focus on the GBA region, with particular focus on the city of Zhuhai, which is a coastal city located in the southeast of the GBA megalopolis and is mainly dominated by plains and low hills with small elevation variations. As a typical representative of the GBA cities, the SWS characteristics in Zhuhai were remarkably altered during the recent decades, which is in line with the rapid urbanization processes during this period. In Zhuhai, meteorological measurements have gradually become increasingly denser. Thus far, there are 98 automatic meteorological observation stations, and 12 of them have been under operation over 10 years.

The primary goals of this study are: (1) to integrate the long-term and high-quality wind observations (both wind and direction) to analyze the SWS trend at a local scale; (2) to quantitatively assess the effects of roughness change on SWS variations with the help of the OMR method, which excludes the atmospheric circulation effects; (3) to introduce a fan-sector method to quantitatively investigate the urbanization effects on roughness change. Ultimately, this study is expected to provide an alternative approach for researchers to investigate urbanization effects on meteorological elements.

## 2. Data and Methodology

### 2.1. Data

There are 84 automatic meteorological observation stations (AMOS) across Zhuhai. Anemometers equipped in each AMOS are installed at a height of 10 m above the surface in order to alleviate the influence of surface boundary layer. Five-minute SWS measurements are collected and then used to calculate the hourly, monthly, and annual means. Note that despite a gap fraction of 0.5% due to instrument damage, communication disruption, or other reasons, the quality of the measurements can be generally guaranteed for the standardized management from the provincial institution. Prior to the acquisition, the data experienced four-level QA/QC procedures, including quality control during data collection, real-time quality control, non-real-time quality control, and manual quality control [28,29]. In this study, we first selected AMOS that have been under operation for more than 10 years. Referring to previously proposed criteria [30,31], we checked the integrity and internal consistency of the data from these stations. After filtering out a subset of stations, we checked the data for homogeneity by using a subjective visual inspection method. The inhomogeneity of wind observation records is mainly manifested in the form of sudden increases or decreases, which may be caused by meteorological changes in the surrounding environment possibly due to site migration. Observational records at qualifying stations must account for 85% of the study period to avoid excessive missing. Finally, we selected 12 AMOS within the period of 2006–2019. Figure 1 illustrates the locations of AMOS sites, which consist of 11 inland sites and 1 island site (i.e., GS station).

The wind speeds during 2006–2019 were also extracted the from the 6 hourly ECMWF latest reanalysis ERA5 [32] at a high resolution of 0.25° × 0.25° to accurately represent the background climatic changes. The data excluded the surface observations during assimilation and fitting, and therefore are less affected by land surface change. The data were first interpolated into the measurement sites using bilinear interpolation and then temporally averaged to derive the daily, monthly, and annual means.

The NPP-VIIRS nighttime light remote sensing data (hereafter referred as nighttime light index, NLI) [33], obtained from the National Oceanic and Atmospheric Administration (NOAA), were used in this study to acquire the detailed urbanization information. These global monthly nighttime light data are a combination of cloud-free images, which were taken by the Suomi-NPP satellite using its Visible Infrared Imaging Radiometer Suite (VIIRS) at an altitude of approximately 824 km from the earth’s surface in polar orbit. The VIIRS offers a significant increase in sharpness and sensitivity over the DMSP satellite detectors used in the past to capture nighttime light images worldwide. The monthly NLI data are available from between 2013–2019, which corresponds to the accelerated development stage of urbanization over the study area. Therefore, this time span is sufficient to illustrate the urbanization process.

Besides the conventional sounding, wind profiler radar is regarded as the most direct way of monitoring upper-air winds both with good accuracy and continuity. To reveal the decadal change of upper-level circulation, the wind speed measurements were also obtained from TWP3 Wind Profiler Radar located near the PS station. The radar has a detection height between 100–3000 m at a temporal resolution of 5 min and is thus less affected by the surface characteristics. In this study, hourly measurements of 700 m wind speed during 2013–2019 were selected for the low data gap (around 25%) and were then averaged into monthly means for use.

### 2.2. Methodology

#### 2.2.1. Mann-Kendall Test

The trend of SWS was analyzed using the Mann–Kendall (MK) test [34,35], which does not require the samples to follow a normal distribution and is less affected by the outliers. The MK test can not only statistically evaluate whether the interested variable has a monotonical trend over time but can also indicate whether the trend is significant or not.

The statistic *S* is defined as:(1)$$S=\sum_{k-1}^{n-1}\sum_{j-1}^{n}sgn\left(x_{j}-x_{k}\right)$$
where $x_{j}$ and $x_{k}$ are the sequential sample values in a sample of size *n*, and
(2)$$sgn\;\left(x_{j}-x_{k}\right)=\left\{\begin{array}{c} 1\;,\;x_{j}-x_{k}>0\\ 0,\;x_{j}-x_{k}=0\\ -1,\;x_{j}-x_{k}<0 \end{array}\right.$$

For *n* ≥ 8, *S* is approximately normally distributed with the mean *E*(*S*) = 0 and its variance is given by:(3)$$VAR\left(S\right)=\frac{1}{18}[n\left(n-1\right)\left(2n+5\right)-\sum_{p-1}^{g}t_{p}\left(t_{p}-1\right)\left(2t_{p}+5\right)]$$
where *g* is the number of tied groups and $t_{p}$ is the number of observations in the *p*-th group. The standardized statistics $Z_{MK}$ for the MK test is calculated as follows:(4)$$Z_{MK}=\left\{\begin{array}{c} \frac{S-1}{\sqrt{VAR\left(S\right)}},\;\;\;\;\;S>0\\ \;\;\;\;\;\;\;0\;,\;\;\;\;\;\;\;\;\;S=0\\ \frac{S+1}{\sqrt{VAR\left(S\right)}},\;\;\;\;\;S<0 \end{array}\right.\;$$

Overall, a positive (negative) value of S indicates an increasing (decreasing) trend. The trend is insignificant if $Z_{MK}$ is less than the standard normal variate $Z_{\alpha /2}$, where *α%* is the significance level.

#### 2.2.2. Observation-Minus-Reanalysis (OMR) Method

Kalnay and Cai [18] proposed a method named OMR to quantitatively evaluate the impacts of urbanization on the trend of surface air temperature. To a certain extent, the ERA5 reanalysis data used in this study were able to represent the characteristics of local climatic change caused by anthropogenic activities, and thus could be used as a reference sequence for local urban climate research [32]. In other words, the change information of local meteorological elements could be separated from the background of global climate change by subtracting the reanalysis (i.e., ERA5) from the observations (i.e., in situ measurements).

In detail, the hourly wind is first decomposed into the zonal component *u* and the meridional component *v*. Then, the hourly OMR wind speed $s_{OMR}$ and direction $d_{OMR}$ are calculated as follows:(5)$$s_{OMR}=\sqrt{\;{\left(u_{obs}-u_{era}\right)}^{2}+{\left(v_{obs}-v_{era}\right)}^{2}}$$
(6)$$d_{OMR}=mod\left(180.0+arctan2\left(u_{obs}-u_{era},\;v_{obs}-v_{era}\right),\;360.0\right)$$

Finally, the monthly and annual OMR wind speeds are calculated based on hourly results.

#### 2.2.3. Fan-Sector Method

The wind speed measured via AMOS is largely influenced by the upstream surface roughness. In order to quantitatively explore the relationship between SWS and surface roughness, the AWOS site is regarded as the center of a circle, which was equally divided into eight 45° sectors, namely, N~NE, NE~E, E~SE, SE-S, S~SW, SW~W, W~NW, and NW~N, respectively (Figure 2).

The upstream fan-averaged NLI for one AMOS in the *s*-th sector within the range of *d* km is given by:(7)$$NLI_{s,d}=\frac{1}{m_{s,d}}\sum_{i=0}^{m_{s,d}}nli_{s,d,i}$$
where *s* = 1, 2, …, 8, *d* is the upstream distances (3, 5, 10, and 15 km in this study), *m* is the number of grids in *s*-th sector in a circle with a radius of *d*, and $nli_{s,d,i}$ is the NLI value on *i*-th grid (*i* = 1, 2, …, m).

Moreover, given a certain distance from the circle center (i.e., the location of the measurement site), the wind speed (after using the OMR method) and corresponding upstream fan-averaged NLI are used to calculate the Pearson correlation coefficient. This method is called CNOF (correlation of averaged NLI and OMR in a fan-sector).

## 3. Result

### 3.1. SWS Variations

Regardless of wind direction, we first focused on the overall trend of SWS. Figure 3a shows the inter-annual variations of monthly SWS during 2006–2019. Overall, the SWS shows an obvious declining trend with a rate of −0.53 m s^−1^ per decade, which is significant at the 95% confidence level. The rate of decrease in Zhuhai is larger than most of the documented rates in other regions [4,5,6,7], since the study area is one of the fastest developing areas in China during the past decades.

Figure 3b shows the decadal trend of monthly SWS at the 12 measurement sites. Note that all inland sites present an evident decreasing trend ranging from −0.22 to −1.00 m s^−1^ per decade, and all of them are statistically significant at a confidence level of 95%. Among the inland sites, the PS site achieves the largest decline slope (−1.00 m s^−1^ per decade), followed by the JZG site (−0.92 m s^−1^ per decade). However, the island site (i.e., GS site) almost exhibits no change, consistent with the background trend over the GBA (Appendix A). It is mainly attributed to the limited development on the island.

The wind frequency is calculated as follows:(8)$$F_{i}=\frac{C_{i}}{C}$$
where $F_{i}\;$ is the frequency of *i*-th grade wind during a stationary period, $C_{i}$ is the count of observed *i*-th grade wind, and C is the total count of observed wind. The definition of the wind speed grade follows the standard of the Beaufort wind scale [36]. Figure 4 illustrates the inter-annual variations of annual wind frequency ranging from grade 1 (light air) to 6 (strong) during 2006–2019. Overall, the weak winds (especially for the second grade) dominate the frequency distribution, and changing trends vary among different wind grades. In particular, the frequencies of both first and second grade winds have been observed with an obvious increasing trend, with the former increasing from 0.18 to more than 0.3 during 2006–2016, and the latter surging since 2006 and reaching 0.44 in 2019. The frequencies of stronger winds have decreased gradually since 2010, with the third (fourth) grade peaking in 2007 (2006) and declining since then. In other words, the long-term decline in SWS in Zhuhai has been mainly caused by the decrease in the frequency of gentle and moderate winds. However, despite the overall SWS decline, the almost invariant frequency of the strong wind could provide an important implication that the risk of the extreme wind remains in the context of global warming, which should be under consideration for both disaster prevention and building construction.

### 3.2. Roughness Change Impacts on SWS

We employed the OMR method to compute the wind speed affected by the surface roughness change. Figure 5 shows the inter-annual variations of averaged SWS from anomalies of observations (OBS) and reanalysis (ERA5), as well as from OMR methods. As compared to the ERA5, both OBS and OMR present significant downward trends (95% confidence level), with the latter (−0.4 m s^−1^ per decade) accounting for 75.5% of the former (−0.53 m s^−1^ per decade). It also indicates a limited change of the background circulation over Zhuhai during recent decades, which has been clearly revealed from both the ERA5 reanalysis (Appendix A) and the observed 700 m radar measurements (Appendix A).

In addition, the declining trend of OMR is also varying among different stations (Figure 6). The OMR trends of all sites are negative and most of them range from −0.2 to −0.7 m s^−1^ per decade. In particular, the PS site presents the largest declining trend at a rate of −0.90 m s^−1^ per decade, followed by HQ (−0.65) and JZG (−0.64) sites. In addition, the declining trends for JD and GB stations are insignificant, which is possibly attributed to the limited degree of the urbanization or environmental uncertainties.

### 3.3. Correspondences between Urbanization and Roughness Changes

In this study, the NLI index is used to represent the urbanization characteristics [37]. Figure 7 displays geographic distributions of the NLI over Zhuhai in 2013 and 2019. It can be seen that the areas with the high NLI are extending and the NLI values are increasing from 2013 to 2019. In the case of 2019, the NLI presents an obvious spatial variation. The east part is typically regarded as the main urban area, corresponding to a high value of 30–45 for NLI, with the maximum of about 50 located in the coastal areas. The coastal area of the west part is also characterized by a high value of NLI ranging from 20 to 40. Moreover, there are several scattered town canters in the central region, with a magnitude range of 15–30 for NLI.

Based on the CNOF method introduced in Section 2, the influence distance for each station is set to 3 km, within which the NLI is averaged over the upstream fan-sector. As a result, the samples are collected four times per day during 2012–2019. Figure 8 displays the variations of OMR according to NLI with the bin width of 5. It is shown that the OMR peaks at a value of 2.4 m s^−1^ when the NLI is small (0–5). Then, the OMR gradually reduces to 1.9 m s^−1^ with the NLI climbing to 40 in company with a moderate oscillation, indicating a clear negative correlation between the SWS and urbanization. However, there is an obvious increasing trend of OMR since the NLI is larger than 40, which corresponds to a high level of urbanization, and could be related to the increased trend of the observed SWS during recent years (Figure 3a). Further discussion about the reversed trend has been conducted in the conclusion.

The annual averages of OMR and upstream fan-averaged NLI are then calculated to explore their relationship at the annual scale. As a result, there are 12 (years) × 7 (stations) samples in total for OMR and NLI, with their correspondences shown in Figure 9a. Similarly, the OMR decreases as the NLI increases with a correlation coefficient of −0.24 at a confidence level of 90%. The slope is 0.01, indicating that the increase in NLI by 1 could lead to reduction of OMR by 0.01 m s^−1^ per decade.

However, the distribution range of OMR is relatively wide when the NLI is low. The reason is that the sparseness of the architectures would restrict their effects on the SWS, and thus including the information of surface roughness effects on SWS would eventually lead to a weakened correlation. Note that there is a wide spread of NLI less than 10 over the western part of Zhuhai (Figure 7), which is typically regarded as the lower (or below) degree of urbanization in previous studies (Appendix A) [38], and could largely degrade the correlation. Therefore, when only making the fitting for the samples of NLI greater than 10, the negative correlation increases to −0.36 (*p* < 0.01), which validates our above assumption.

Furthermore, to identify the effects of the influenced distance on the correspondences between NLI and SWS, Figure 9b–d shows the increases in the upstream distance used in the CNOF method to 5, 10, and 15 km, respectively. Note that a distance of 3 km (or larger) is required to perform the fan-sector analysis due to the limit of the NLI resolution (~0.5 km). It is shown that the correlation becomes weaker with the increase in the influence distance. However, as compared to the control experiment (3km), the negative correlation increases to −0.51 (5 km), −0.76 (10 km), and −0.44 (15 km) when only accounting for the samples of NLI greater than 10. The possible reason for the increased correlation is that the dense and tall buildings in the urban area could lead to the formation of the downstream far-field low wind speed (DFLWS) zone at a certain distance. Relevantly, Tsang et al. [2] used the wind tunnel to test the surface wind environment around tall buildings and identified an obvious DFLWS zone under different experimental configurations, which is related to the reattachment of the vertical recirculation behind the building and is also affected by the strength of the horizontal recirculation.

## 4. Discussion

The CNOF method has been established to validate the basic assumption that there is a remarkably negatively correlation between NLI and OMR in the fan-sectors, with the former representing the urbanization characteristics, and the latter denoting the SWS excluding the atmospheric circulation effects. Despite the comprehensive consideration as being possible, there are still some errors or inaccuracies listed as follows.

First, NLI is a reasonable, but not the best, proxy for the urbanization characteristics. For example, some tall buildings may also be characterized by weak light. Zhang et al. [37] systematically assessed the usage of nighttime light data in the representation of urbanization and found that NLI could account for as much as 93% of urbanization changes despite some overestimation. Second, despite the extensive use of the OMR method, some errors could be generated when using the ERA5 reanalysis data at a local scale. Nevertheless, the changing trend of OMR is still statistically significant, favoring the application of this method. Third, the SWS may include some noises arising from anthropogenic activities or other interference, which could be partially eliminated after temporal average. In order to avoid artificial skill and fully examine the robustness of the conclusion, additional OMR methods are also performed on each individual site over Zhuhai (Appendix A). It is shown that nearly all sites present a significant decreased trend of SWS in the context of the small variations of background circulation. Fourth, urbanization is a combination of the increasing building heights in existing built-up areas and the increasing number of new buildings around the periphery. This is not specifically addressed when only taking the upwind extent of the sector into account.

## 5. Conclusions

In this study, the effects of urbanization on SWS in the GBA megalopolis, as exemplified by the city Zhuhai, is investigated with the use of long-term automatic meteorological measurements, ERA5 reanalysis, and nighttime light data.

Based on the analysis of the observational data, we found that the SWS decreased at the rate of −0.53 m s^−1^ per decade, and the roughness change could account for as much as 75.5% of the decreased via the OMR method. Moreover, both SWS and OMR at most of the inland sites presented significant downward trends. In particular, the declining trends of SWS ranged from −0.22 to −1.00 m s^−1^ per decade, which could be largely accounted for by the change of OMR.

Furthermore, the CNOF method was proposed to quantitatively analyze the correspondences between urbanization and roughness changes. It is shown that when the upstream distance is set to 3 km, the correlation between OMR and the upstream fan-averaged NLI reaches −0.24 (statistically significant), which could be increased to −0.36 when only the samples of NLI greater than 10 are taken into account. In summary, there was a good sign of significant relevance between NLI and OMR by using the CNOF method, which offers an additional option for assessing the urbanization effects on meteorological elements.

Although there is a controversy on the key factors responsible for the long-term SWS change, our study has well demonstrated that it is the rapid development of urbanization, rather than the change of large-scale circulation that is mainly responsible for the change. Our results have determined the significant SWS decline over the GBA megalopolis, which may provide important implications for researchers to assess the urbanization effects at a local scale.

An obvious increasing trend of SWS at a global scale has been confirmed since 2010 [26], probably determined by ocean–atmosphere oscillations. Note that a noticeable increasing trend of SWS has also been identified at Zhuhai since 2017. Due to the weak variations of ERA5 reanalysis over the GBA megalopolis (Appendix A), this increasing trend could not be reasonably attributed to climatic change. A probable reason is that when the urbanization reaches a certain high-level, the tall and dense buildings could enhance the SWS possibly due to the “canyon effects” or other reasons. It has been partially demonstrated by the reversed relationship between NLI and SWS when NLI is larger than 40 (Figure 8). Consequently, ongoing research has been conducted to apply the large-eddy simulation (LES) to model the wind conditions under the different building configurations, which is of great importance to illustrate the mechanism of correspondences between SWS and urbanization, and could also provide implications on the research for other aspects (e.g., pollutant dispersion and outdoor human comfort).

Additionally, the radar measurement is only used to represent the variations of background circulations and could further be utilized to enhance its connection with the SWS and urbanization over the GBA in subsequent studies. For example, LiDAR technologies are capable of capturing wind characteristics and profiles in complex terrain regions [39] and could also provide valuable information in research on urban ventilation, air pollution dispersion, sea breezes, heat island and flow disturbances caused by tall buildings due to urbanization [40]. Moreover, future research could employ the wind profiler radar to reveal and understand the wind characteristics of the urban boundary layer (UBL) under different circumstances [41].

## Figures and Tables

**Figure 1 ijerph-19-03194-f001:**
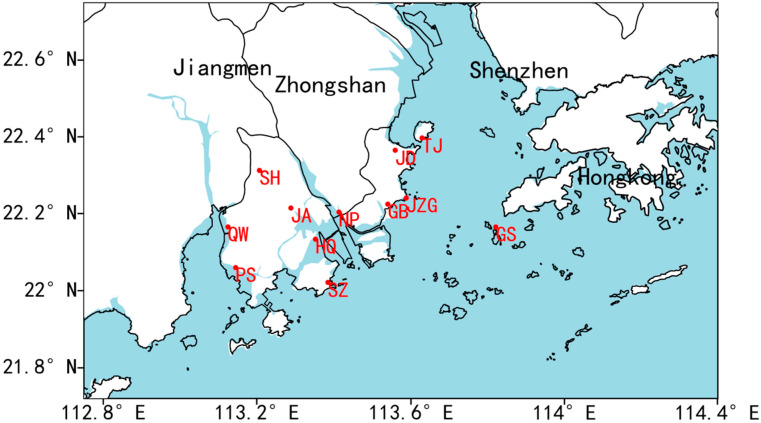
The map of the GBA megalopolis with the red circles denoting locations of measurement sites over Zhuhai (GS: GuiShan Island; QW: QianWu; JD: JinDing; SZ: SanZao; NP: NanPing; PS: PingSha; JZG: JiuZhouGang; TJ: TangJia; GB: GongBei; JA: JingAn; SH: ShangHeng; HQ: HongQi).

**Figure 2 ijerph-19-03194-f002:**
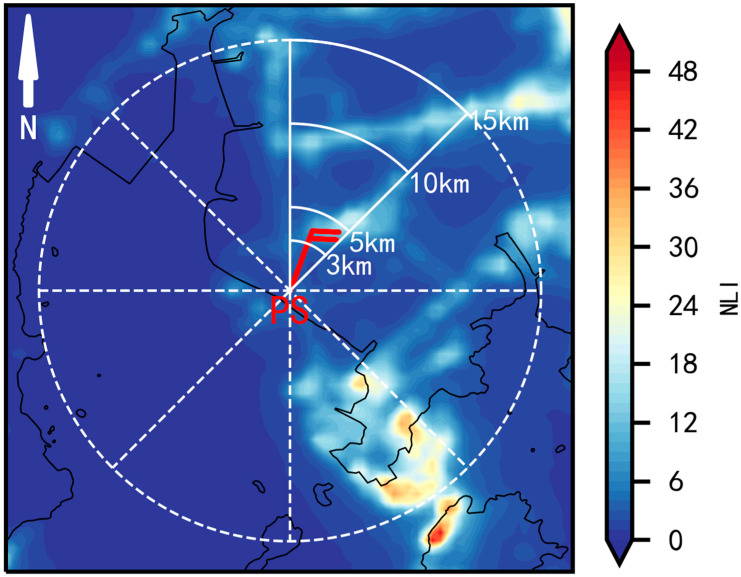
Schematic diagram of the fan-sector method.

**Figure 3 ijerph-19-03194-f003:**
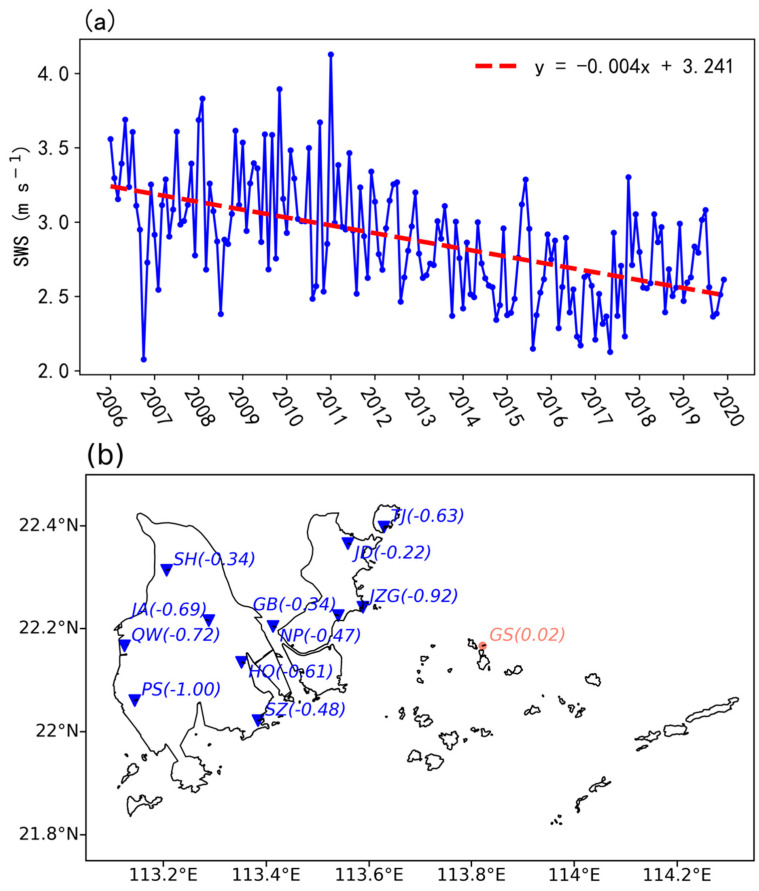
(**a**) Inter-annual variations of monthly SWS (m s^−1^) averaged over Zhuhai during 2006–2020, with the black dashed line denoting the fitted line. (**b**) Geographic distribution of the decadal trend (m s^−1^ per decade, in brackets) of monthly SWS over Zhuhai.

**Figure 4 ijerph-19-03194-f004:**
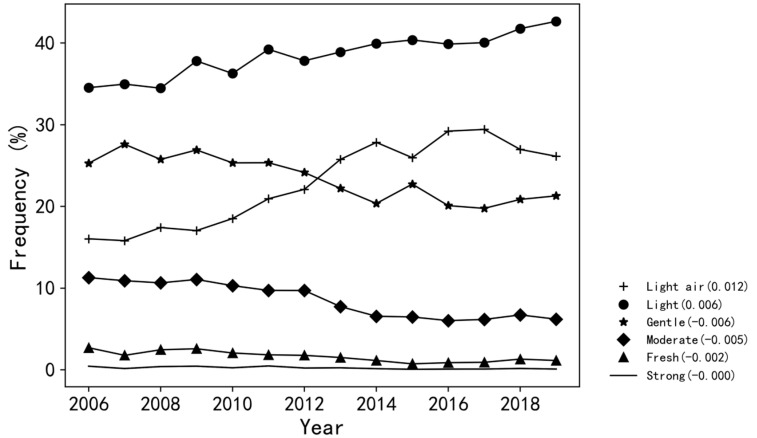
Inter-annual variations of observed annual wind frequency (%) ranging from grade 1 (light air) to 6 (strong) during 2006–2019.

**Figure 5 ijerph-19-03194-f005:**
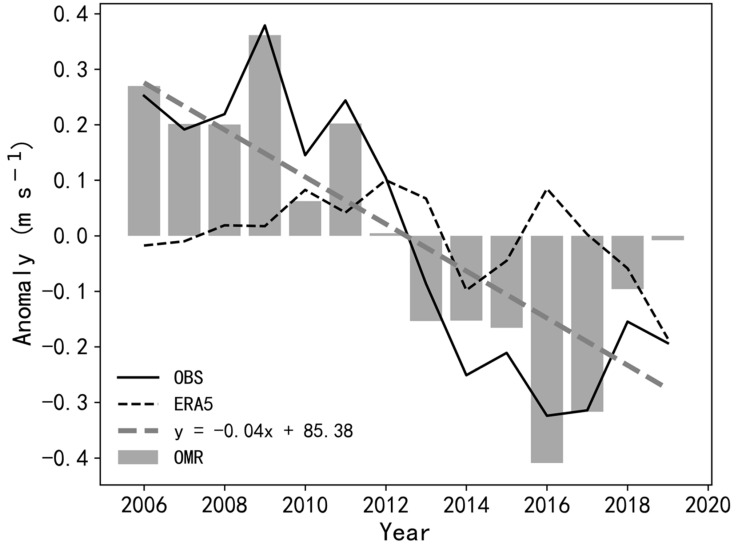
Inter-annual variations of SWS (m s^−1^) averaged over Zhuhai from anomalies of observations (OBS) and reanalysis (ERA5), as well as from OMR methods.

**Figure 6 ijerph-19-03194-f006:**
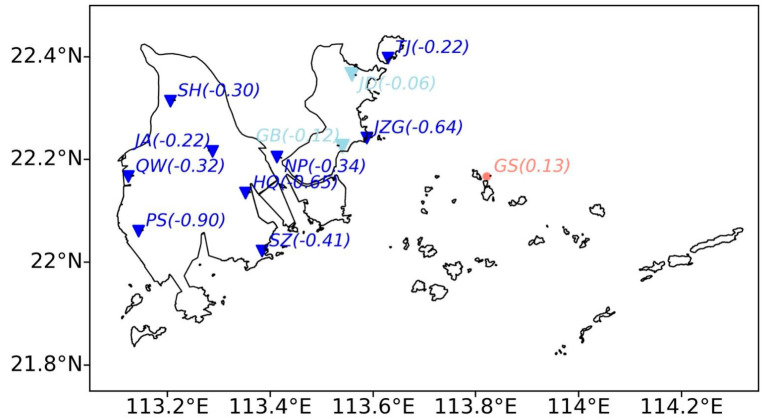
Geographic distribution of the decadal trend (m s^−1^ per decade, in brackets) of monthly OMR over Zhuhai.

**Figure 7 ijerph-19-03194-f007:**
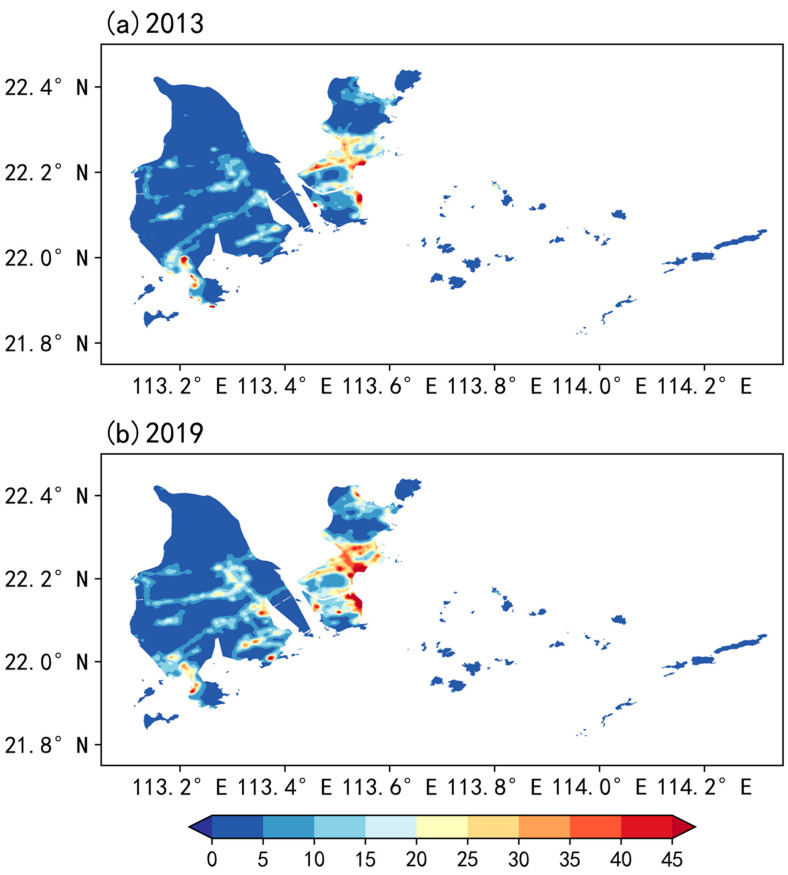
Geographic distribution of the NLI over Zhuhai in (**a**) 2013 and (**b**) 2019.

**Figure 8 ijerph-19-03194-f008:**
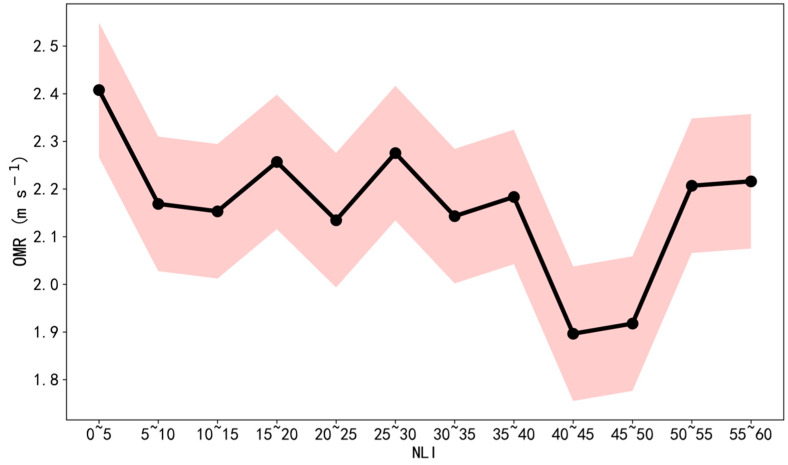
Variations of OMR (m s^−1^) according to NLI, with the shadings denoting the standard deviations.

**Figure 9 ijerph-19-03194-f009:**
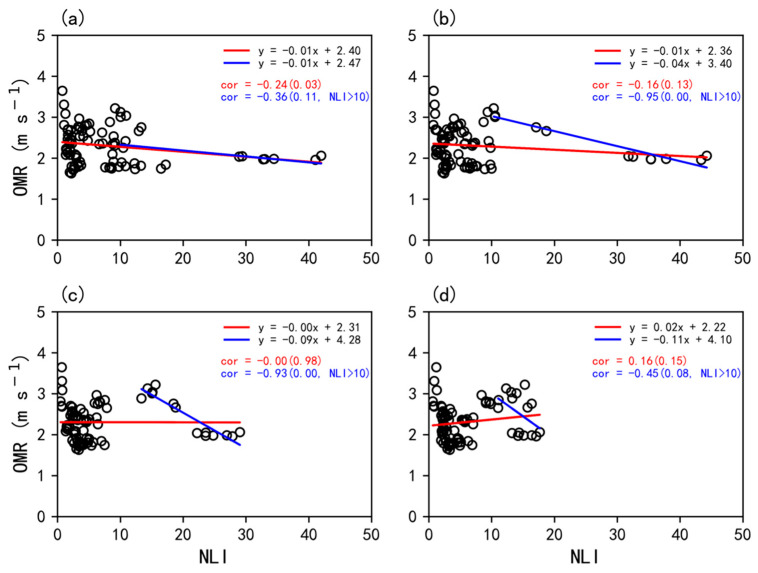
Scatter diagrams of annual OMR (m s^−1^) against NKL using the fan-sector method when the upstream distance is set to (**a**) 3, (**b**) 5, (**c**) 10, and (**d**) 15 km, with the red and blue lines denoting the fitted lines for all samples and samples of NLI greater than 10 only, respectively.

## Data Availability

The AMOS and Radar measurements are not publicly available due to the demand for data confidentiality. The NLI remote sensing data and ERA5 reanalysis are downloaded online from https://eogdata.mines.edu/products/vnl/ (accessed on 24 February 2021) and https://cds.climate.copernicus.eu/cdsapp#!/dataset/reanalysis-era5-single-levels?tab=overview, respectively (accessed on 14 June 2018).

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
