# Peer review of "Urbanization Effects on Surface Wind in the Guangdong–Hong Kong–Macao Greater Bay Area Using a Fan-Sector Method"

_ijerph, 2022, doi:10.3390/ijerph19063194_

Round 1
Reviewer 1 Report
Primary concerns are lack of citing current research in urbanization effects on SWS (ie Kalnay, E., Cai, M.: Impact of urbanization and land‐use change on climate. Nature. 423, 528–531, (2003). 345 https://doi.org/10.1038/nature01952)
see Yang, P., Ren, G., Yan, P. et al. Tempospatial Pattern of Surface Wind Speed and the “Urban Stilling Island” in Beijing City. J Meteorol Res 34, 986–996 (2020). https://doi.org/10.1007/s13351-020-9135-5
as well as more current research that indicates overall SWS is increasing since 2010 due to climate change see Zeng, Z., Ziegler, A.D., Searchinger, T. et al. A reversal in global terrestrial stilling and its implications for wind energy production. Nat. Clim. Chang. 9, 979–985 (2019). https://doi.org/10.1038/s41558-019-0622-6
Comments in more detail:
The main issue lies in how the author has framed the relevancy and merit of the article. I believe that the core component, the “urbanization effects on surface wind in 2 GBA megalopolis using a fan-sector method” presents a novel approach to identifying what would essentially be a wind shadow caused by the roughness of urbanizing areas. However, as I pointed out, the introductory section lines 35-64:
Please note that nearly all of these citations are averaged across rural and open areas and most are before the noted INCREASE in SWS caused by Climate Change identified in 2010 and onward. This was initially reported in Yang, P., Ren, G., Yan, P. et al. Tempospatial Pattern of Surface Wind Speed and the “Urban Stilling Island” in Beijing City. J Meteorol Res 34, 986–996 (2020). https: //doi.org/10.1007/s13351-020-9135-5 and I gave that reference as a starting point for updating the introduction:
“Based on meteorological measurements, a number of studies have revealed the de-35 crease of surface wind speeds (SWS) during the past decades over China (Mcvicar et al. 36 2010; Ying et al. 2010; Fu et al. 2011; Guo et al. 2011; Yue et al. 2020), America (Buckley 37 and Weber 2006; Pryor et al. 2009; Monahan et al. 2011; Greene et al. 2012), Europe (Najac 38 and Terray 2009; Smits and Knnen 2010; Azorin-Molina et al. 2016), and other regions 39 around the world (Pirazzoli and Tomasin 2003; Buckley and Weber 2006; Mcvicar et al. 40 2008; Brázdil et al. 2010; Vautard et al. 2010; Garmashov et al. 2011). For example, the 10th 41 and 90th percentile of the hourly mean wind speeds over the UK have declined signifi-42 cantly from 1980 to 2010 (Earl et al. 2013). Turkey has experienced a decrease of the annual 43 average wind speed at a rate of 0.14 m s-1 decade-1 during 1975-2006 (Dadaser‐Celik and 44 Cengiz 2014). A significant decrease of the wind speed during 1974-2002 was detected over France, particularly in the Mediterranean area (Najac and Terray 2009; Najac et al. 46 2015). Similarly, a noticeable decrease was also found in Canada and North America for 47 the period of 1945-1995 (Teller 2010) and 1979-1999 (Ge et al. 2010) respectively. For China, 48 a decline of the SWS has also been reported in many studies. Fu et al. (2011) found that 49 the SWS over China showed a distinct decline trend from 1961 to 2007. Ying et al. (2010) 50 reported that the annual mean SWS declined at a rate of -0.12 m s-1 decade-1 from 1956 to 51 2004. You et al. (2010) noted that both surface observations and the National Centre for 52 Environmental Prediction (NCEP) reanalysis revealed the significant decreasing trend of 53 the SWS over the Tibetan Plateau at a rate of -0.24 and -0.13 m s-1 decade-1, respectively. 54 Mcvicar et al. (2010) identified a pronounced decrease of SWS over central China during 55 1960-2006 with a higher rate at higher elevations than lower elevations.”
Again we see that although the research cited is after 2010 the datasets are prior to the shift
“Based on in-situ 56 measurements, Wu et al. (2016) revealed a distinct decrease of SWS at a rate of -0.23 m s-1 57 decade-1 over the Eastern China Plain (ECP) region during 1980-2011.”
And again, although the cited article is published in 2017 the data is prior to 2010:
“Recently, Wu et al. 58 (2017) analysed the change of SWS around the world during 1973-2005, and found that 59 the most significant decreases occurred in Central Asia and North America at a rate of -60 0.11 m s-1 decade-1, followed by Europe, East Asia, and South Asia with mean linear trends 61 of -0.08 m s-1 decade-1. 62”
The result of the perspective has led the author to this oversimplified and mostly erroneous statement, because we know that near-surface winds are probably determined by ocean–atmosphere oscillations, rather than by vegetation growth and/or urbanization:
“The change of SWS is under the influence of large-scale circulation and surface 63 roughness.”
My suggestions to the author are to reapproach this section and cite more current findings and reword in light of this more recent SWS phenomena. Additionally, I would ask the author to asses the datasets that were used to ensure they accurately reflect those findings. While I realize that that is a major change in the article, I believe the author’s novel fan-sector approach has value and when presented in light of more current findings could benefit from publication.
Author Response
We thank you for providing these valuable comments on the introduction. In the revised manuscript, we have reorganized the introduction through incorporating further improvements especially as you suggested. The major modifications on the introduction in the revised manuscript are to: 1) rephrase the same text as the abstract, and enhance the connection between surface wind and anthropogenic activities; 2) include the description of climatic change; 3) include the description of reversal of terrestrial stilling (i.e., SWS decline) at the global scale identified in 2010 and onward probably due to the internal decadal ocean–atmosphere oscillations; 4) emphasize the different roles of urbanizations and climatic change in determining the SWS variations at different scales (from local to global) and regions (from rural to urban); 5) cite current research in urbanization effects on SWS; 6) remove the less relevant or similar references and put more focus on the research progresses on GBA megalopolis; 7) improve the connection between the topic of this study and the scope of the journal (i.e., public health & environmental degradation).
Please view the revised manuscript for the modified introduction.

Reviewer 2 Report
This manuscript is well-written and presents novel and original research. It was a pleasure to read. My only comment is that the quality of the graphics could be slightly improved and a compass added to Fig. 1.
Author Response
Please view our detailed responses in the attached file.

Reviewer 3 Report
This is a well-written paper and I do not have many comments on this. The subject is not novel, but the authors have presented their topic, analysis and write-up very well. Therefore, I recommend a publication of this MS.
I have three minor comments here:
Line 28-34: The sentences are the same as in the abstract. You need to rephrase them. Do not repeat the same sentences in the abstract or conclusions.
Line 101: "and annual means"
Line 168: "fastest" not fasted
Author Response

(The authors gave the same response as above.)

Reviewer 4 Report
The authors have explored how urbanization could affect surface wind speeds in GBA megalopolis (mainly at Zhuhai), and they adopted OMR analysis together with fan-sector method for conducting analysis. The topic itself is important, however the current manuscript is too short and "clean", i.e., more technical details, explanations, together with potential extensions, and the combination of measurement datasets with LiDAR measurements and equipment should be mentioned and explored within the manuscript. The followings show many important points that the authors must provide in-depth explanations and technical details before re-submission:
Major issues:
(1) Lines 36-62: this part should become much shortened, because many of these discussions have no direct connections with the main objectives of this manuscript - the introduction of manuscript should get more focused on GBA megalopolis
(2) Lines 55-56: How does this related to the elevation discrepancy within the GBA megalopolis? How's the situation at GBA?
(3) Line 129: only data of October from 2013-2019 were used for analysis - that might not be trustworthy - abrupt structural / urbanization changes might have taken place in some October, should take annual mean data, despite the uncertainty caused by cloud fraction / cloud coverage.
(4) Section 2.2: Methodology should be hugely expanded - for example, adding in equations, data analysis, description of data selection etc. Also, how the MK test is applied, how do you categorize "Fan Sector" and make good use of it etc.?
(5) Lines 153-155: should also provide some information about how NLI is calculated / defined.
(6) Section 3.1: GS site - how can you justify it is a good representation of background climatic conditions? You will need more comparisons before reaching such conclusions.
(7) Line 181 (caption of Figure 3): does not match with satellite data (2013-2019), and does not necessarily imply the years when Zhuhai experienced serious urban expansion and development. Thus, the graphs should be re-plotted, which focus on the years whether Zhuhai really experienced urban development and expansion.
(8) Lines 184-196: What is the social implication / scientific implication? Can we see or review the urbanization effects from the temporal plot?
(9) Lines 202-203: Need to add some explanations of the data handling process, and data retrieval / data removal process of OMR algorithm.
(10) Figure 7: not much practical use, should also include the corresponding figure in 2016 or earlier, to clearly illustrate the urbanization effects that have taken place among all years.
(11) Lines 235-237: seems contradictory - may not be totally true, need to mention some special cases (fluctuating trend), for example, when NLI = 30-35 or 40-45
(12) Lines 259-261: Is there any previous work supporting the methodology - of taking samples / fitting samples of NLI greater than 10 only? How is the proportion of data samples with NLI less than 10?
(13) Lines 271-272: The reason may not be an appropriate and sufficient reason, the authors should also combine with other environmental uncertainties, data errors etc. Further, actually your Sections 4.1 and 4.2 can be combined.
(14) Lines 283-285: This will cause huge experimental or statistical errors, since the amount of data used for analysis is insufficient, or too less. Again, why October is a representative month? Please explain.
(15) Section 4 in overall: A case study at different place of Zhuhai should also be included into this manuscript - with corresponding spatial discrepancy in terms of urbanization clearly spell out within the case study, then repeat the methodology again, and see the corresponding statistical connection between NLI and SWS. Some conclusions can be significant from such case study.
(16) Conclusion: The conclusion needs to be extended, currently, it is too limited, and may not consist of scientific impacts.
(17) LiDAR technologies: In Sections 4 and 5, the authors should also connect with the use of LiDAR technologies for conducting surface wind measurements, as well as obtaining vertical wind profile, in urbanized areas, for the purpose of validation or long term assessments - relevant papers are as follows, the authors should focus more on data integration for conducting wind profile assessments, and the use of other well available equipment like wind LiDARs.
(i)https://www.sciencedirect.com/science/article/pii/S2212095521001140?via%3Dihub
(ii) https://www.mdpi.com/1996-1073/13/19/5135
(iii) https://www.mdpi.com/2072-4292/11/21/2522
(iv) https://iopscience.iop.org/article/10.1088/1757-899X/276/1/012004
Minor issues:
(1) Lines 30-33: the authors should highlight meteorological changes, climatic variation etc.
(2) Lines 35-41: no need cite that many sources for each country, just 2 sources will be good enough here.
(3) Lines 53-54: Tibet may not be related to the spatial scope of this manuscript.
(4) Line 76: can provide a bit of elaboration of OMR method here.
(5) Lines 86-87 - Goal 1: do you mean "wind" observations? if so, please state it more clearly, wind direction, wind speed, wind shear, vertical wind profile or?
(6) Line 101: mean wind speed, or wind direction, or both?
(7) Lines 102-104: What kind of QA/QC procedures have it undergone? Any relevant official governmental documents from Zhuhai?
(8) Line 110 - "changes in the surrounding environment" : in what manner? structural changes, or meteorological changes?
(9) Line 126: do you mean only 2013-2019 NLI datasets of Zhuhai are available, or you simply extract these years?
(10) Lines 145-147 - just to clarify: Do you mean NCEP data represents local climatic changes all over the years?
(11) Lines 165-168: should remove some unnecessary citations.
(12) Lines 170-177: Where are these sites (please provide their full names, longitudes, latitudes)?
(13) Figure 3(b): Please provide full name of all these stations, both inland and island stations, and their labels in the figure.
(14) Line 185 (the equation): move to centre of the line., also please add i = 1, 2, 3..., 6.
(15) Lines 211-213: Need some extra information regarding the geographical positions and meteorological conditions of these sites.
(16) Line 220 (Caption of Figure 6): Caption should include the full name of these stations.
(17) Figure 8: Is there any error bound of these numerical figures? Please also include these error bounds in the figure.
(18) Line 267: What do you mean by "variations of NLI become weak"?
(19) Line 289: What is the "average algorithm" technique that you have applied? Explain briefly.
Minor grammatical errors are found as well, for example:
(1) Line 16 and the rest of this manuscript (please change all of them): decade^-1 can be changed to "per decade" to avoid confusion, since there has already been a ^-1 in wind speed unit.
(2) Line 18: that excludes
(3) Line 22: correlation can reach 0.40 (in negative manner)
(4) Line 50: distinct declining trend
(5) Line 72: that " the increase of"
(6) Line 73: "should mainly be responsible for a decrement of"
(7) Line 82: for "climate change assessments"
(8) Line 88: "that excludes"
(9) Line 142: "climatic" changes
(10) Line 158: Schematic "diagram" of
(11) Line 164: "It is of no surprise that the rate of decrease in Zhuhai is larger..."
In overall, the authors should have addressed all aforementioned issues - and expand the scope and coverage, as well as technical details of the current manuscript. The presentation currently is clear, but lack of scientific rigor and profession, therefore please add in all technical details, equations for wind speed retrieval / assessments, as well as the statistical metrics being adopted for comparison. The authors can also conduct a case study at selected place of Zhuhai or Guangdong Province, so that the numerical details can be reviewed in a better manner. Future data integration (e.g., LiDAR measurements) for wind speed retrieval etc. should also be properly acknowledged and addressed in Discussions and Conclusions of this manuscript, which will result in better scientific insights.
Author Response

(The authors gave the same response as above.)

Reviewer 5 Report
The paper deals with quite a narrow topic, but it is important for understanding of wind change effects at microscale of a Chinese megapolis.
Despite the manuscript makes a contribution to the field, there are some concerns that should be addressed before the paper is ready for proceeding with publication.
- The title must be completely reworded. “An investigation” is redundant as a first word. The GBA abbreviation is not common and must be written completely.
- The connection between the topic of the paper and journal title must be expressed in a more explicit manner. Currently, it seems like you study a sole effect of wind change due to urbanization and do not make a proper link with possible harm to the public health, the issues of environmental degradation and so forth. This connection must be shown in all the sections of your study.
- The formula used to test the appropriate hypotheses must be explicitly written in the Methodology section. There are a lot of variations even in widely spread statistical tests, such as Mann-Kendall, so it is important to show the methodology in full.
- The conclusion is too short and very technical. What are the main outcomes from your analysis? How it should be used for future research and practice? What are the consequences of wind speed decrease if projected for next 30—50 years?
- I would also recommend an extensive editing of English language and style.
Author Response

(The authors gave the same response as above.)

Round 2
Reviewer 1 Report
The author has added additional text that recognizes the increasing global SWS since 2010 and included that in the conclusions. The Abstract, however, has not been updated to reflect this addition. Considering most readers will decide to dig into the article based upon the Abstract, it is recommended that the author revise the abstract to reflect the revised and updated conclusions.
Author Response

(The authors gave the same response as above.)

Reviewer 4 Report
The modified version of this manuscript looks good, and the authors have addressed most of the comments that we previously raised, we appreciate the authors' efforts a lot. However, there are still some modifications to be made / points to add before acceptance of this manuscript, namely the followings:
(1) Point (11) of previous review: The wording here is still not too accurate, as it may lead to confusion -"In general, the OMR decreases as the NLI increases when the NLI is less than 40, indicating a robust negative correlation between the SWS and urbanization.". Since we can only see fluctuating trends here and the curve bounces up and down, therefore it may not be appropriate to put this statement in. Further, the authors are suggested to put emphasis on the range of NLI = 35-55. Here, negative correlation when NLI is small does not mean much.
(2) Point (13) of previous review - here, the new reason "that the dense and tall buildings in the urban area could have an important influence on the downstream SWS at a certain distance." sounds more appropriate. But could the authors provide more experimental results, or some supporting evidence in this context? i.e., more elaboration can be provided.
(3) Point (17) of previous review - it's good that the authors have include more information here. We understand that win could provide the vertical profile of wind, although it may be beyond the scope of this study, it's really worthwhile and good to connect with wind LiDAR technologies (e.g., pulsed Doppler wind LiDARs and wind farms) and outline it as a future plan or extension based on your current study, possibly in the Discussion or Conclusion sections of this paper.
LiDAR technologies have been widely applied in modernized city, for example, for observing the upwind, downtown and downwind vertical wind profiles, and its connection with urbanization effects in Hong Kong (at Supersite of HKUST (an university of Hong Kong), Hong Kong Observatory and Cape D'Aguilar); for capturing wind characteristics and profiles in complex terrain regions of Japan- by placing LiDAR on a wind farm near seashore with downwind turbines, thus the change of wind profile / wind speed could be accounted by local or city-wise urbanization effects.
Relevant Papers and Links:
https://www.sciencedirect.com/science/article/pii/S2212095521001140?via%3Dihub
https://www.mdpi.com/1996-1073/13/19/5135
https://www.mdpi.com/2072-4292/11/21/2522
We hope the authors can connect current study with these previous findings, especially in the context of the Guangdong-Hong Kong-Macao Greater Bay Area, which is close to both Hong Kong and Japan.
(4) Minor Comment No. (7) - about QA/QC procedures: Could the authors also include the confidence level / the reliability of these available measurements in Macau? This will be very important.
(5) Minor Comment No. (18) - "spatial variations of NLI become less obvious due to the limited resolution (~ 0.5 km)" What is the range of variation of NLI?
(6) Indeed, in the last paragraph of our previous report report, we have put down this comment: "Future data integration (e.g., LiDAR measurements) for wind speed retrieval etc. should also be addressed in Discussions and Conclusions of this manuscript" - therefore, corresponding comments and suggestions have been made in Point (3) of this review report (shown above). We think this will be very important for the future of this discipline, for monitoring changes of wind speed / direction measurement at local or city-wise manner.
Other comments of the previous review report have been clearly addressed, and we appreciate the authors' efforts and time for enhancing the quality of this manuscript.
Author Response

(The authors gave the same response as above.)

Reviewer 5 Report
I am satisfied with the corrections made by the authors. Good luck with publication.
Author Response
We thank you for recognizing the value of our work.